# The Superior Visual Perception Hypothesis: Neuroaesthetics of Cave Art

**DOI:** 10.3390/bs11060081

**Published:** 2021-05-26

**Authors:** Per Olav Folgerø, Christer Johansson, Linn Heidi Stokkedal

**Affiliations:** Department of Linguistic, Literary and Aesthetic Studies, University of Bergen, 5007 Bergen, Norway; photo@linnheidi.com

**Keywords:** neuro-aesthetics, visual thinkers, perception, cognition, neanderthal genes, language

## Abstract

Cave Art in the Upper Paleolithic presents a boost of creativity and visual thinking. What can explain these savant-like paintings? The normal brain function in modern man rarely supports the creation of highly detailed paintings, particularly the convincing representation of animal movement, without extensive training and access to modern technology. Differences in neuro-signaling and brain anatomy between modern and archaic Homo sapiens could also cause differences in perception. The brain of archaic Homo sapiens could perceive raw detailed information without using pre-established top-down concepts, as opposed to the common understanding of the normal modern *non-savant* brain driven by top-down control. Some ancient genes preserved in modern humans may be expressed in rare disorders. Researchers have compared Cave Art with art made by people with autism spectrum disorder. We propose that archaic primary consciousness, as opposed to modern secondary consciousness, included a savant-like perception with a superior richness of details compared to modern man. Modern people with high frequencies of Neanderthal genes, have notable anatomical features such as increased skull width in the occipital and parietal visual areas. We hypothesize that the anatomical differences are functional and may allow a different path to visual perception.

## 1. Introduction

The extremely detailed Cave Art from the European Upper Paleolithic, with its accurate depiction of motion, is a mystery, as it is not as symbolic as so much other Cave Art. The ability to perceive, imagine and retrieve the necessary visual information from previous episodes of viewing live animals is normally far beyond the capabilities of modern man, except for some with savant talents. These ancient people did not have cameras and were unlikely to take notes without portable media such as paper; they relied solely on raw perception and memory. Prehistoric cave paintings show evidence of abilities that are typically rare and often regarded as pathological in modern man. We will use the term Cave Art to mean figurative, detailed art performed predominantly as an “indoor” activity, in opposition to Rock Art that is predominantly symbolic and exposed outdoor.

Our analysis relies on analogical reasoning comparing painters in the past with savant painters in the present. Cave Art demonstrates an ability to perceive anatomical details and motion similar to how Savant Art shows savant perceptual ability and memory today. The evidence is multi-faceted and documented not only in the products but also in the brain anatomy and, recently, genetics, as neanderthal genes associated with perception and cognition have been revealed in populations outside of Africa and we are only beginning to understand the functions and evolutionary preservation of these gene alternatives. That the ancient artists possessed cognitive and anatomical features associated with visual thinkers and some modern-day savants poses the question of whether they also had more access to raw perception, enabling the accurate perception of both anatomical details and motion, and less access to the top-down processing we associate with language processing (cf. [1,2,3,4]). Neanderthal gene variants associated with visual perception have been found in modern man, and people with high frequencies of Neanderthal genes have notable anatomical features such as increased skull width in the occipital and parietal visual areas [5]. Some findings suggest that inhibiting top-down processing can improve abilities that rely on accurate perception. Could the loss of superior visual perception be the price for language?

## 2. Materials and Methods

Our materials rely on analyzing several sources, from savant-like features of Cave Art to modern accounts of savant-like skills in autists, and the discovery of archaic genes linked to anatomically enhanced visual areas. We begin by presenting evidence of savant-like representations in Cave Art. We will move on to discuss savant-like visual perception in autistic savants, before discussing recent genetic evidence for the involvement of an archaic visual perception system in the creation of Cave Art.

### 2.1. Savant-Like Representation of Details

Details such as the accurate multiplication of lines for the representation of feet in fast running reveal a high temporal dynamic. How is it possible to remember the fine details of an animal without the animal present? The painters may have had an extraordinary aptitude and particular training in memorizing animals, possibly serving some ritualized function. A modern analogue is the memory studies of Maguire et al. [6] who compared the memory capacity of London taxi drivers to that of London bus drivers. Brain scanning using functional magnetic resonance imagining (fMRI) showed an enlarged posterior hippocampus in the taxi drivers compared to the bus drivers, who have a less varied, and more predictable, route to navigate. Malafouris [7] stresses that our brain is in continuous development and is in strict adherence to culture and our material environment. Roberts [8] states that “we have never been behaviourally modern” to emphasize the continuous role of material culture in the hominid evolution.

The brain of archaic Homo sapiens shows significant differences to that of modern man. Archaic Homo sapiens was presumably tuned to perceive more raw details (bottom-up perception) and use raw memory undisturbed of conceptual thought structures (top-down control). This allows representational abilities for detail when generalization is limited. Such abilities turn on in modern man when the left anterior temporal lobe is temporarily *shut down* by transcranial magnetic stimulation [2]. A more information-rich perception and detailed memory puts higher demands on the raw processing ability, which is consonant with larger brain volumes in archaic Homo sapiens [9].

We define raw perception as perception that is closer to sensory perception, undisturbed by higher-order concepts and top-down processing, for example linguistic processing. Individuals that rely more on raw perception are predicted to be less disturbed by conflicting top-down information and would be less likely to let such information change memories when they are not linked to higher-order categories. We will come back to this in the discussion in Section 4.

The creative capacity in Cave Art is reminiscent of drawings by savant autistic children; a famous example is the case of Nadia [10]. Ramachandran and Hirstein [11] draw on Nadia’s extraordinary capacity in this single field as an example of brain modularity, i.e., where a single brain region is hyperactive, while most other areas are more or less dysfunctional, or disturbed by the hyperactive areas. The case of Nadia suggests that creative capacity is located in the right parietal cortex and either compensates for, or causes, reduced left anterior temporal lobe activity, with consequences for speech [11,12].

Were the painters of Upper Palaeolithic autistic savants? Humphrey [12] compares drawings by Nadia with paintings of the Upper Palaeolithic era and raises the question whether there is a connection between Nadia’s impaired language abilities and her drawing abilities, and whether the same was the case for the savant-like cave painters. A significant point in Humphrey’s reflections is that: “A person not only *does not need* a typical modern mind to draw like that (in the caves) but *must not have* a typical modern mind to draw like that. Then the cave paintings might actually be taken to be proof positive that cave artists’ minds were essentially pre-modern” [12] (p. 171, our italics). Given that the cave painters did not have access to modern equipment and training it is difficult to otherwise understand how they created accurate details, both anatomically and in dynamic motion. This will be discussed in Section 2.2.

The cave painters were certainly not autistic savants according to modern diagnostics. Their creative skills, however, show that their visual, cognitive, and perceptive abilities far exceeded what we today will call normal. A savant-like visual capacity was normal, as evidenced by enhanced visual areas of the brain, and hence non-pathologic, for archaic Homo sapiens. In a recent article, Masataka [13] draws on the term *neurodiversity* when referring to two of his commentators [14,15]: “[…] cognitive as well as emotional properties characteristic of developmental disorders such as autism spectrum disorders (ASD) are not necessarily deficits but fall into normal behavioural variations exhibited by humans” [13] (p. 122). Spikins stresses that “Upper Palaeolithic art in South–Western Europe is dominated by often extraordinary realistic and naturalistic depictions of animals, both on cave walls and in portable art. A number of elements of this art, such as a highly realistic detailed figural representation, a focus on parts (with drawings often overlapping) and a remarkable visual memory from what can only have been limited opportunities to note details of dynamic animals are found in common with [modern] autism” [16] (p. 14).

“The Savant Hypothesis” of Fabricius suggests a neuronal basis for the savant’s ability to record all details from the primary visual areas V1 and V2 [17]. This is consistent with Koshino et al.’s [18] findings, referred to in Mottron et al. [19], that while non-autistics during perception get an activation of left frontal regions (roughly corresponding to areas associated with language and reasoning), “autistics exhibit mainly extrastriate activation, consistent with their optional use of a more perceptual mechanism” [19] (p. 1389). Frith [20], via Pring [21] (p. 224), similarly stress the over-activation of early sensory processing in autism, i.e., a bias towards extraordinary abilities of perception, contrary to later processing areas.

### 2.2. Animals in Motion

The savant-like paintings of the Upper Palaeolithic show attention to details, as well as visual memory, related to the topics of movement, savant-like perception and the *Enhanced Perceptual Functioning* (EPF) in the savant brain [12,19,22,23]. According to EPF there is a “skewing of brain activation towards primary and associative areas in autistics” [23] (p. 39) while non-autistics use higher centres and top-down control over perception [17].

Features of the animals depicted in prehistoric art include the representation of naturalistic movements, animals running at high speed, or involved in fights, particularly noticeable in the paintings of the Chauvet cave (France; 32,000–36,000 BP). The paintings are sketchy and strongly dynamic. The interested reader will find many pictures at places such as the Bradshaw Foundation, especially the Lion Panel of the Chauvet Cave.

Animal movement in Palaeolithic paintings, is represented in examples from Chauvet, the Sanctuary of Les Trois Fréres, Lascaux, as well as many other sites [24,25], for example, see [24] for a comparison of animal postures from different sites. The artists “… arrived at two processes for breaking down movements, the first by the superimposition of successive images, the second by the juxtaposition of successive images […]. Representations take two forms: either by the addition of a second version, more or less complete, of the part of the body concerned, or by the multiplication of barely sketched contours (lines) around the head or legs, which generates a sort of dynamic flux.” [25]. Dynamics may take the form of multiplied legs, as in a bison of Chauvet [25] (p. 318, their Figure 2). The animal tableau found in the Sanctuary in Les Trois Frères (Ariège, France; c. 14,000 BP), like the much older Chauvet, shows a strong dynamic movement of the animals. The representation here is *engraved* onto the wall and shows how painters in the Palaeolithic era could make images appear naturalistic, as animals under great excitation and movement.

The representation of movement “… is an essential factor in parietal art. For the artists of the Upper Palaeolithic, movement was an integral part of the process of understanding the animal” [26] (p. 118). The morphology of the cave walls, such as niches and recesses, was a dramatic scenery in the position of figures; edges suggest body shapes, virtually rendering drawings into sculptures, seemingly floating on the rock [27] (pp. 210ff.).

The incorporation of the rock surface makes the imagery fluctuate between the two- and three-dimensional [28] (p. 58). This is helped by the light from torches, which create the drama. “One can presume that these figures, which appear inert, came to life in their [the artist and the spectator’s] minds, moving according to the fluctuations of the lighting on the irregular volumes of the wall, by changing the angle of observation (anamorphosis) or according to the distortion of perception caused by alteration of consciousness [visual hallucinations]” [26] (p. 121).

The animal representations in the caves vary from an isolated deer standing in profile to a herd of animals under enormous speed, fighting, running from a predator, or attacking their prey. You get the impression that the affective brain took control over the cognitive parts and released the hand from cognitive restraints. It is no longer a calm description of how a rhinoceros, a lion, a bison etc. looks; it describes the animals as they appear in nature, describing how they act, and the tremendous energy they release as they change from still images to those of a herd. Astonishingly, the way they depict the animals in motion is surprisingly accurate. Eadweard Muybridge in 1886 was the first to film how horses run, and it was confirmed that the so-called primitive cave man painted their leap almost correctly with an error rate of 46.2% vs. 65.2% in modern paintings [29], making the animated animals even more impressive.

### 2.3. Perception in Modern Autists, Savants and Visual Thinkers

#### Autism

Grandin [30] gives an inside perspective of a visual thinker on the autistic spectrum. She stresses that autistic spectrum disorder is highly variable in its outcome and predictions, and the criteria for diagnoses have changed, and are now divided into several subclasses, each with their own internal variance. Several aspects stand out as common denominators, such as sensory disorders and cross talk between senses that causes much distress to the individual. It could be near impossible to integrate an experience from many sensory modalities: some accounts show that an attention to one sensory modality inhibits another, for example being virtually blind while listening, or vice versa deaf while attending to vision. The increased attention and retention of details in some individuals is of particular interest to our hypothesis.

Grandin [30] (p. 184), citing Rimland [31], states that about 10 percent of people with autism have savant skills, and some of those have specialized skills in enhanced visual perception and visual thinking. She characterizes herself as a primarily visual thinker who has used her visual thinking to overcome many obstacles that face autistic people. One problem for the extreme visual thinker could be related to bottom-up thinking, where detailed imagery makes generalization harder to accomplish because of the high-specificity of an image compared to general symbols: there is nothing in the visually perceived that stands out as the most significant. Non-autistic thinkers may be extreme on the other end of the spectrum, in that top-down general concepts create expectations that guide our thinking and even our perception and make it hard to see details or alternatives outside of our conceptual framing.

The enhanced perception of details fits well with the finding that “even though language deficits are a diagnostic criterion for autism, at least one aspect of language may actually be *enhanced”* i.e., picture naming, [32] (p. 1398). Moreover, (with reference to [33]) “the ‘disorder’ of autism may constitute not a cluster of deficits, but rather a set of relative strengths and weaknesses across various domains” [32]. The high coincidence of savantism and autism relies on the enhancement of lexical/semantic memory, a neurocognitive strength characterizing the condition [32]. This and the enhanced visual perception and retention of details, as well as fast perception capable of detecting details in movement points toward a special kind of visual thinker.

Enhanced perception may result from many different paths. Research has shown hyper-connectivity in the autistic brain [34], larger visual areas [35], and increased neuronal excitability, for example from an imbalance of inhibitory GABA (gamma amino butyric acid) signaling and excitatory glutamate [36,37,38]. The enhancement of motion perception is also confirmed in autism [39].

### 2.4. Archaic Homo Sapiens and Genetic Links to Raw Perception

Are high-level concepts, mediated through language, in conflict with a heightened perception of detail, as seems the case in modern autistic savants?

“[S]avants have privileged access to lower level, less processed information before it is packaged into holistic concepts and meaningful labels. Owing to a failure of top-down inhibition, they can tap into information that exists in all our brains but is normally beyond conscious awareness. This suggests why savant skills might arise spontaneously in otherwise normal people and why such skills might be artificially induced by low frequency repetitive transcranial magnetic stimulation (rTMS)” [4] (p. 1400). The rTMS was directed toward the left frontotemporal lobe and the left anterior temporal lobe [2,3,4], which would temporarily inhibit those areas. rTMS resulted in improved drawing skills in normal healthy persons, who showed improvement in these skills lasting up to 45 min after stimulation. Low frequency rTMS temporarily inhibits neural activity in a localized area of the cerebral cortex [4]. This reduces the activity in networks responsible for putting detailed information from perception into pre-established concepts, which enables access to details directly drawn from raw perception and resembles the suggested condition of the brains of savants, but in completely normal adults with temporarily knocked-out brain areas associated with language and top-down control.

Using another protocol, Chi et al. [40] confirmed the results [4], using transcranial direct current stimulation (tDCS). This method showed that a reduction in anterior temporal lobe activity resulted in *increased visual memory*: “Only participants who received *left* cathode stimulation (decrease in excitability) together with *right* anodal stimulation (increase in excitability) showed an improvement in visual memory. This 110% improvement in visual memory was similar to the advantage [of] people with autistic memory” [40]. The strong coherence between the results of Snyder [4] and Chi [40] provides strong evidence in favor of the thesis that autists have an extraordinary access to raw perception. This demonstrates that the pathways are present, and that regulation of pathways, such as downregulations of top-down control, may explain savant skills and, by analogy, the acute vision in the archaic cave painter.

There is evidence for the presence of genetic material “shared by Neanderthals, Denisovans, and by a small fraction of non-African modern human. [Material that is known to play a role in] GABA-mediated neurotransmission in the central nervous system, in particular in neocortical and hippocampal neurons.” [41] (p. 12). This *ingressed haplotype* plausibly “result in alterations of synaptic plasticity” [41] (p. 12).

Gobet et al. [42] propose that decreasing top-down conceptual processing and increasing the role of low-level perceptual processing leads to boosted creativity. What blocks our access to the raw perception? One explanation is, as we have seen, that inherent concepts are driving us, and these concepts are a hindrance for finding new and better solutions. This is the so-called *Einstellungseffect* [43,44], which is a block for better solutions in, for example, professional chess players. Eliminating the *Einstellungseffect* will give us better access to raw perception [42] (p. 174). Could the boost of extreme creativity and memory in archaic man taking place about 40kyBP, as evidenced by cave paintings, be explained by the archaic brain possessing a natural ability to imprint raw perception into memory, undisturbed by top-down conceptual processing? If such modes of perception were common in the population at the time, it would hardly be a *pathology* in that culture.

The Savant Hypothesis explains the difference between savant perception and normal perception as differences in neural inhibition of information-carrying neurons [17]. According to this, feedback resonance from higher-order cognition and acquired concepts activate the inhibitory system of neurons. The result is a compression of incoming signals, preventing over-stimulation of the brains’ cognitive apparatus. People with normal perceptual abilities can compress the neuronal message to a prototypical representation necessary for understanding and incorporating it into higher cognition. One example is linguistic concepts that abstract from a rich information to the significant and relevant features possible to express in words. The savants, lacking in this ability to compress the signal, thus receive information from all the neurons. Compression would also perform “error correction”, which would affect the sensitivity of detection, thus preventing false “positives” i.e., hallucinated features, as weak signaling and inhibited neurons would not send their piece of information at all.

The explanation proposed by [17] fits well with observations of people with acquired Savant Syndrome, such as those who are artificially induced by transcranial magnetic stimulation of the frontotemporal cortex, as discussed above. These higher areas normally feed inhibitory stimuli back to V1 and V2, resulting in silencing their neurons. Without this negative feedback, the “cognition reverts back to the primary sensory cortices and become concrete and detailed in character. Individuals with such injuries demonstrate more detail awareness at the expense of devastated cognitive abilities” [17] (p. 263). The autistic brain is (genetically) strongly governed by increased excitation (glutamate) and reduced inhibition by GABA [36,37,45,46,47] resulting in altered neuronal resonance patterns in reciprocal connections. Moreover, there are long distance innervation from monoamine-releasing neurons (noradrenaline, serotonin, dopamine) and acetyl cholinergic neurons, stimulating, inhibiting, or modulating the interaction with other neurons [47,48].

In fact, inhibition and feedback are constrained in a different balance in the savant versus the normal brain, and by analogy (we suggest) in the archaic versus the modern brain (Figure 1).

The ability of the savant brain to perceive every detail is reminiscent of the working memory capacity of young chimps demonstrating superior memory of numeral sequences [49]. Chimps trained in numeral recognition, and particularly the young ones, were able to remember long sequences of numbers, such as 1-2-3-4-5-6-7-8-9, and remember their correct place on a screen after seeing them for 210 ms (milliseconds), before blocking them by white squares (even after being distracted for 10 s). The correctness for picking the numbers in order was 80% [49] (their Figure 2). The limit of a possible saccadic movement of the eyes is close to 250 ms. “This means that this condition does not leave subjects enough time to explore the screen by eye movements” [49] (R1005). The chimps could also handle sequences with missing elements without loss of performance. We find this extreme ability in chimps’ working memory a very interesting corollary to the savant brain, and a demonstration of a superior effect of raw perception that is present in a related species with a common primate ancestor, suggesting that archaic man may have had similar visual abilities, where modern man does not. Possibly this loss is a price to pay for analytic, linguistic thinking.

In suggesting that the archaic brain had a different balance, including features resembling those found in modern savants, we *do not* say that archaic Homo sapiens sapiens were savants according to modern pathological terms. However, some features deriving from the archaic genomes may show up in modern man and alter her/his perception and visual memory, and this alteration shares traits with modern autism. The mere possession of archaic genes, however, shows that this does not reliably predict autism, as up to 4% of populations outside Africa have genes associated with Neanderthals [50].

#### Autism and Brain Size—Evolutionary Aspects

Our modern brains have recently evolved towards smaller volumes. People with autism seem to follow a different development: the increased brain size in autism is contrary to the general evolution of Homo sapiens. The results of Sacco et al.’s study [51], p. 249) demonstrate that autism is associated with an atypically connected and often overgrown brain. Similarly, “an abnormal excess number of neurons in the prefrontal cortex” [52] (p. 2001) is observed. The brain size in autism is increased within the first years post-natal and then, to a differing degree, reduced toward normal at adulthood [53] (their Figure 2), [54].

ASD subjects have increased dendritic spine densities for pyramidal neurons in the frontal, temporal, and parietal cortices [55]. There are numerous alterations of the connectivity during the first years after birth. The amount of connectivity differs as to whether it is local or long range. The local hyper-connectivity in ASD stands in stark contrast to the hypo-connectivity between distant cortical locations [56] (p. 154). “Functional imaging studies have shown a reduction in coordinated activity between distant cortical regions and an absence of the top-down modulation of early sensory processing that is found in typically developing individuals” [55] (p. 123). This shows an anatomical correlate to the assumed reduction of top-down processing in autism [17], with hyper-perception of raw information as a consequence [4].

During evolution, the brain size has shrunk in the last 25,000 years, losing a volume “the size of a tennis ball” [57] via [9], a process possibly associated with the spread of agriculture or domestication of animals [58] and the self-domestication of our own species [9]. The development of general intelligence is not only a result of brain size, which may be driven by evolutionary forces and only indirectly related to intelligence, such as neoteny (the selection of young features), and an adaptation or exaptation as a reuse of existing structures [59,60]. Falk [61] via [62] suggested that brain size and increased convolution could start as an adaptation caused by a need to cool the brain, emerging from bipedal gait and increased temperatures due to climate change. Hofman [62] emphasizes the importance of brain connectivity, the proportion of white to gray matter, and that the human brain is far from its optimal processing capacity, although the optimum would make natural birth extremely complicated, because of the narrow birth channel.

Early man, Cro Magnon, as well as Homo neanderthalensis, had larger brains than in modern humans, whose brains are reduced in volume as a result of neoteny and domestication [9,57,63]. Hood [9] sums up research that shows similar effects of domestication of animals compared to the recent shrinkage of the human brain. Domestication may originate in neoteny, which is the retention of phylogenetically early features, which can be thought of as cute features. This is, however, not necessarily the same as selectively preferring non-aggressive phenotypes, which we associate with domestication, even though the effects may be similar.

Domesticated animals and humans have in common that they work best by recruiting help from others that they know may have solutions to a problem. This requires a different type of intelligence than that necessary for individual survival, and it is a more sophisticated intelligence employing information that is not immediately available, or even hidden from direct perception, such as reasoning about the mental states and intentions of others: the so-called Theory of Mind (ToM) network [64].

The autistic brain shows significant differences in ToM [64], and recent research has demonstrated that people with ASD have a significantly reduced activation in the ToM-associated brain areas in response to direct gaze [65] compared to normal controls. Insight into the state of mind of others through a theory of mind is a hypothetical interpretation in mental terms of opaque behaviors, i.e., that the individual uses reasoning that can be expressed in propositional logic to infer insights into the mind of others. The alternative is embodied simulation that “rejects both the [reliance on abstract symbolic representations] and the standard forms of simulation theory that depend primarily on explicit simulations of the other’s internal state and that require explicitly taking the perspective of the other, by relying on introspection” [66]. Gallese suggests “…embodied simulation as a mandatory, nonconscious, and prereflexive mechanism that is not the result of a deliberate and conscious cognitive effort aimed at interpreting the intentions hidden in the overt behavior of others […]” [66]. Embodied simulation thus suggests a pre-verbal, pre-rational, and non-predicative model in which *the I* will be reflexively mirrored in the other, and *vice versa*, all within the same brain-body system [67,68].

The mirror neuron system (MNS) therefore makes the observations related to a theory of mind fully plausible, even without abstract reasoning about others. Individuals with ASD suffer a dysfunction of their MNS, impeding the sensory/motor and the emotional mirroring mechanisms, preventing a person’s ability to understand other people’s emotions, resulting in a lack of empathy [69,70]. This means that the normal understanding of the intention intuitively attributed to gestures and face expressions is more or less absent in people with ASD. In general, social relationships alters in persons with ASD, and “[s]ocial consciousness is assumed to engage cortical areas, including the superior-temporal-sulcus, the temporoparietal junction, and the medial PFC (prefrontal cortex), mostly in the right hemisphere” [71] (p. 176). The dysfunction of autism has, however, not led to an evolutionary elimination of ASD even though it leads to pathological problems in areas that are important to humans. The autistic condition in Asperger’s syndrome may be associated with an extraordinary expertise for subjects such as engineering and mathematics [22] (p. 6). There is no reason to consider all facets of ASD as solely negative characteristics in a Darwinian conception of evolution [13]. During the development of the human mind, however, toward more and more domestication, and in pace with a shrinking brain volume, a person with ASD seems to conserve traits of an archaic larger brain. Evolution normally replaced or outvoted these archaic traits, replacing them with networks catering for abstraction, language, and socialization. The cost of lost access to an archaic configuration is the lost ability for detailed raw perception and access to visual memory. The modern brain utilizes an extensive pruning of local connectivity. In ASD, this pruning is delayed to such a degree that “the differences between autistic and typically developing individuals mirror the differences between younger and older typically developing individuals” [72] (p. 4).

### 2.5. Sudden Boost of Creativity: What about the Neanderthals?

Bednarik [73] has argued against the idea of an invasion from Africa to Europe of gracile modern humans that suddenly replaced the robust, archaic humans. Instead, he proposes an extended co-existence of robusts (sapiens neanderthalensis) and graciles (sapiens sapiens), and that they were extremes within the same human population. This is confirmed in studies of the human genome, which in Eurasia includes 1–4% of genes deriving from Homo sapiens neanderthalensis [50] (p. 721), showing that their co-existence included interbreeding. The technological culture of archaic humans (including sapiens neanderthalensis) allowed their spread over Eurasia and Australia, and their cultures “were sufficiently advanced to support skilled navigation of the open sea […]” implying an extensive use of tools [73] (p. 2). Hardy et al. have recently published evidence of Neanderthal textile fiber technology demonstrating advanced cognition with deep implications for behavior [74].

Evolution favored gracile features, and in essence, a preservation of gracile juvenile features (neotony) that moves at different rates in males and females. There is an unexplained acceleration in gracilization in females 40,000 years before present. “The process has continued to the Holocene, and reduction in both dimorphism and robusticity is also still active in human evolution today” [73] (p. 11). A modern brain organization for perception may result from a brain reorganization associated with self-domestication [73] (p. 11) [9].

### 2.6. Volume of Visual Areas: Sapiens Sapiens vs. Sapiens Neanderthalensis

Recent research is getting closer to pinpointing the mechanisms that result in an increased neocortical area. Boyd et al. [75] have shown a biological regulatory mechanism that is selectively different in humans compared to chimpanzees, and the human variant accelerates neocortex development through interactions with multiple genes. Expansion of the neocortex in turn may support many general functions as demonstrated by Stowe et al. [76] in a meta-study of brain areas implicated in language processing as well as other functions such as working memory. The increase in brain size in the evolution of the Homo line, until the more recent shrinkage of H. sapiens’ brain volume, in turn allowed the coordination of larger populations, which promoted communicating through symbols and speech [77,78].

However, cooperating with others is also a source of stress, and it is not hard to imagine that when populations reached a critical mass the evolutionary scenario changed into a scenario of self-domestication, through down-regulating stress reactions [9], which may eventually lead to differential reproductive success, which is the driving force of evolution. As the level of aggression lowered in early societies, their potential to grow increased and created a positive feedback loop that gradually resulted in the elimination of robust traits, when these traits were not advantageous, i.e., in domestication. With domestication followed an enhanced cultural evolution, and organization of societies, which coincides with our interest for a visual world as reproduced on cave walls, i.e., what we now call Cave Art. During the Pleistocene period, the two different sub-species, sapiens neanderthalensis and sapiens sapiens, co-existed and interbred [5]. They left some extraordinary artefacts, and some extraordinary genes.

The difference between gracile and robust variants of Homo sapiens is manifested in the outward visible anatomical layout, and in brain anatomy. Pearce et al. states that: “…Neanderthals had significantly larger visual systems than contemporary anatomically modern humans (indexed by orbital volume) …” [79]; Kochiyama et al. found that early Homo sapiens had “relatively larger cerebellar hemispheres but a smaller occipital region in the cerebrum than Neanderthals long before the time that Neanderthals disappeared” [80], and adds that such a neuroanatomical difference in the cerebellum may have caused important differences in cognitive and social abilities between the two species and might have contributed to the replacement of Neanderthals by early Homo sapiens. Oghiara et al., based on a digital reconstruction of the Neanderthal and early Homo sapiens endocasts, conclude that a large occipital region may be a *shared* morphological characteristic between Neanderthal and early Homo sapiens [81] (p. 25). Kochiyama et al., referring to Pearce et al. [79], estimated larger Neanderthal visual cortices based on the orbit size of fossil crania, stating that “(I)n support of Pearce et al., the occipital region was significantly larger in NT (Neanderthal) than in EH (Early Homo sapiens sapiens) in the present study” [80] (p. 5). Recently, Bruner (2021) holds that “a noticeable correlation between orbit size and the size of the occipital lobe suggests that Neandertals might have had a larger occipital cortex, when compared with modern humans” [82]. Significantly, he states that “(i)n this sense, the larger proportion of occipital cortex in Neandertals, if confirmed, should be probably intended as a primitive human condition, and not as a derived Neandertal-only feature” [82].

Carhart-Harris et al. [83] discuss access to *primary consciousness,* as related to the entropy of the brain. They present evidence that heightened states of creativity could be reached in modern man if using a drug (psilocybin), which elevates brain signaling and lowers the control of thought processes. Creativity could be boosted by lowering higher-order control, thus decreasing conceptual processing and emphasizing low-level perception that is not normally available to our awareness; as we have seen above, this was also suggested by Gobet et al. [42] and others. Transcranial stimulation is another alternative as previously mentioned [2,4,40]. It could be that simply avoiding linguistic left-hemisphere activation, through suppressing the use of language, could have similar effects using meditation, or similar techniques, without the use of drugs or machinery.

Bednarik states that “[t]here are only three realistic alternatives to account for the EUP [Early Upper Paleolithic] tool, Cave Art, and portable art traditions: (i) that they are the work of Neanderthals, or of the (ii) descendants of Neanderthals, or of (iii) invading, perhaps genocidal Moderns” [84] (p. 352). The second alternative is harmonious with recent genetic findings of neanderthal genes in modern humans. Green et al. writes: “[W]e cannot currently rule out a scenario in which the ancestral population of present-day non-Africans was more closely related to Neanderthals than the ancestral population of present-day Africans due to ancient substructure within Africa” [50] (p. 722). If there were modern humans in the Upper Paleolithic, it is not at all certain that they had the same genome as present-day people.

One question is why Neanderthal genes are missing in Africa. As is stated by Green, for a long time the paradigm has been that “all present-day humans trace all their ancestry back to a small African population that expanded and replaced archaic forms of humans without admixture. Our analysis of the Neanderthal genome may not be compatible with this view because Neanderthals are on average closer to individuals in Eurasia than to individuals in Africa” [50] (p. 721). Thus, there is every reason to believe that the gracile and robust versions of man were a long distance away from each other for a long time.

Marean [85,86] presented a theory based on the archaeological evidence that dramatic climatic changes making Earth both dryer and colder, took place 164,000 years ago, i.e., about 50,000 to 100,000 years after the appearance of our species. This resulted in a southward movement of sapiens sapiens where they settled near the coast of today’s South Africa; the most studied caves are those at the Pinnacle Point (PP), particularly the so-called PP13B. Those people ate shellfish, rich in proteins, and carbohydrate rich geophytes. The societies were small, only some hundred people, who in fact became progenitors of all later sapiens sapiens. Since these people relied on the sea for food, and originated from a small group of survivors, it is not unlikely that over time, when the population re-established, some aquatic adaptations occurred and much nearer in time than the Aquatic Ape Hypothesis suggests (cf. [87]). Sapiens sapiens lived in these caves from the time of the climatic bottleneck at about 164,000 years BP to about 35,000 BP when the climate ameliorated, and people moved north. This explains why the interbreeding between sapiens sapiens (south in Africa) and sapiens neanderthalensis (cold North) did not happen south of the Sahara. It was, hence, in the period following the migration north that the two sapiens species met each other, interbred, and expanded over Eurasia. As stressed above, Neanderthal genes are present in modern populations [50], and such genes are associated with brain volume expansions that are consistent with expansions of primary visual areas [5,79,80]. If the artists that made Cave Art were not identical to modern humans, but rather archaic humans that included both robust (neanderthalensis) and gracile (sapiens sapiens) individuals, it is reasonable that they kept better adaptions for perceiving and processing visual features and motion.

Recent research has demonstrated that there is a rich ’fauna’ of iconic art in Indonesia (Sulawesi, cave of Leang Bulu´Sipong 4) painted as early as 43–44 thousand years BP. Being the earliest hunting scene known in prehistoric art, animals, and hunters (here: theriantopes) appear here in impressive fast movement [88]. Recently, Finch et al. published the oldest known animal Rock Art in Australia, found in the Kimberley region: “Notably, one painting of a kangaroo is securely dated to between 17,500 and 17,100 years on the basis of the ages of three overlying and three underlying wasp nests. This is the oldest radiometrically dated in situ rock painting so far reported in Australia” [89].

After the advent of representation systems, such as writing, with possible precursors e.g., in earlier cave paintings and symbols, we saw the emergence of coordinated civilizations. In simulation studies, it was demonstrated that the learnability of late acquired language features is affected by changes to the social network, without any obvious change to individual brain capacity [90], supporting the idea that expressions may adapt to our ability to learn and perceive.

The (south west) European Cave Art of the Pleistocene is unique in that it is highly figurative, and almost superhuman in its detailed depiction of animal extremities in motion, as evidenced by a surprisingly low error rate in quadruped walking depictions [29]. Figurative art is lacking in the earlier art by sapiens sapiens from South Africa [91], in Blombos cave.

## 3. Results

*A Testable Hypothesis.* The evidence of larger visual areas in Neanderthals leads to our testable hypothesis suggesting that an archaic visual system, in some respects, was superior to the common visual system in modern humans, given that archaic genes associated with visual perception have been found in some modern people. The problem is to motivate screening for such gene variants. Zeberg and Pääbo [92,93] present methods for screening for Neanderthal genes, as they turn out to be both a major risk factor for conditions related to chromosome 3 [92] and protective against issues linked to chromosome 12 [93] and COVID-19, thus motivating a clinical interest in researching the presence of archaic genes. There are many tests for visual acuity, and visual memory, apart from asking people to draw detailed images. As an example, Conty et al. [94] present a test for visual acuity as measured by perceptual reaction time. A pathway for enhanced visual acuity awaits further research but may involve functional brain imaging (fMRI). Do modern people with enhanced visual areas, and a high proportion of Neanderthal (robust) genes [5,95], have functional advantages in visual acuity and visual memory? Furthermore, is there enhanced visual perception and are the pathways in their brains compatible with raw perception as outlined in this article? Can the balance between excitation and inhibition have been similar in Neanderthals as to what we find in the autistic brain? Hyperexcitation will result in more acute vision (as in autistic savants). Inhibition, excitation and feedback loops are constrained in a different balance in the savant versus the normal brain. Can it, by analogy, which is our postulate, have been similar mechanisms in the archaic versus the modern brain?

## 4. Discussion: Enhanced Visual Perception

We earlier hinted that part of the puzzle is that raw perception has benefits. Higher-order perception is less than optimal in, for example, recall tasks. This has been demonstrated experimentally by temporarily knocking out brain areas associated with higher-order control, as discussed in Section 2.4. There is also other evidence from less drastic manipulations. Wammes et al. demonstrated improved recall of word lists when subjects made drawings rather than writing [96]. One explanation is that when we activate words in the mental lexicon, we also activate closely related words through a process called *spreading activation*. This makes the recall task more difficult as we have to select between close alternatives. Research into *false memories* also shows how malleable memories are to verbal suggestions and recency effects. Loftus demonstrates how memories are reconstructed internally each time they are accessed and integrated with other new information [97]. This reconstruction happens top-down, and a hypothesis is that people who rely more on raw perception may have a more accurate recollection of real experienced events. Morrot et al. demonstrated that a simple manipulation, such as coloring white wine red, activates vocabulary adequate for describing the taste of red wine, and thus changes the perception of the taste of the wine significantly [98]. This is related to cross-modal perception (cf. the McGurk effect), where different senses are integrated to create a perception. In this case, top-down information through vocabulary activated by color may change the taste perception radically. These examples show that without the top-down influence the perception and recall are likely more accurate.

The evidence points towards archaic humans having extraordinary visual acuity and memory for visual details and motion. They were also able to draw realistic figurative images that capture both detail and motion correctly. Pearce et al. [79] demonstrated that Neanderthals had significantly larger orbits than anatomically modern humans. Kochiyama et. al. [80] and Ogihara et al. [81] used digital endocast reconstructions of Neanderthal and early Homo sapiens, showing increased visual cortical areas. Visual brain areas were larger for archaic man than for modern man. Recently, it was shown that modern people with a high proportion of archaic (Neanderthal) genes have enlarged visual areas [5]. In a recent study, “functional connectivity analyses revealed that what is called *primary IPS* (i.e., intraparietal sulcus)—*NeanderScore association* was driven by increased IPS functional connectivity with regions subserving visual processing, but decreased connectivity with regions implicated in the neurobiology of social processing” [93]; our italics). Thus, studying modern visual thinkers, and their association to the NeanderScore, may give us unique insights into the world of the visual thinkers that created Cave Art so long ago. Visual thinkers would certainly have fit into a society that probably had less emphasis on verbal skills. We do not know what people of the Upper Palaeolithic era talked about, but there is a good chance that the artistic skills of highly perceptive people would be as appreciated as those who could tell a good story. Possibly, as today, telling a good story may have been a rare ability.

As we have seen above, 1 to 4% of the genomes of people in Eurasia derive from Neanderthals [50] (p. 721). There is, however, no significant difference in this proportion between Europe and Asia, and there is no evidence of any similar proportion of such genes in Africa. It is estimated that the gene flow was mainly from Neanderthals to modern humans, with no evidence of the reverse direction of flow [50] (p. 721). There is also no evidence for mitochondrial DNA from Neanderthals [50] (p. 710) in modern humans, which is consistent with a low rate of interbreeding. Green et al. provide information [50] (their Table 2, pp. 714–715) about gene differences, showing 78 nucleotide substitutions for the change of protein-coding capacity in modern human genes, while “Neandertals carry the ancestral (chimpanzee-like) state” [50]. Thus, relatively few amino acid changes have become fixed in the last few hundred thousand years of human (sapiens sapiens’ genome) evolution [50] (p. 715). For comparison, in Neanderthals the gene is ancestral. By the same token, Green et al. also state that in sapiens sapiens there are “293 consecutive SNP (snip) positions in the first half of the gene *AUTS2*, where only ancestral alleles are observed in the Neanderthals” [50] (p. 717). Mutations of *AUTS2* (and also in gene *CADPS2*) are implicated in autism [99]. The picture indicates that cognitive traits have been under selective pressure in modern humans, but Neanderthals may have had other genetic traits perceptually as well as cognitively, without this being pathological, and possibly the pathology emerged in modern humans because of selective pressure towards a different, possibly less robust, balance of perception and cognition.

In a recent development, Gregory et al. [5] managed to associate Neanderthal genes with specific anatomical deviations. Their method involves calculating an individual index for known Neanderthal genes and to relate that index to observed changes in skull and brain morphology, assessed per individual using Magnetic Resonance Imagining. Their results show an association between a higher score for Neanderthal genes and a “more Neanderthal-like skull shape […] as well as regional changes in brain morphology underlying these skull changes, specifically in the Intraparietal sulcus (IPS) and visual cortex” [5] (pp. 5ff.). Although this only shows an association between the genes and morphological changes, the changes are consonant with what we know about Neanderthal skull shapes. It is hard to know if the genes directly cause the change, or if they act indirectly through interaction with other genes (many of which are shared between Neanderthal and modern humans). The IPS participates in visual processing and is “particularly critical for tool making” [5] (p. 6). The primary visual cortex “is responsible for the first steps in processing of visual information in the mammalian cortex and feeds into later brain regions in the ventral and dorsal visual processing streams […], with the IPS playing a prominent role in the latter” [5] (p. 6). The role of the IPS and the dorsal visual path, the so called “where system”, is entirely consistent with the suggested enhanced visual perception of detail and motion in Neanderthals and hybrids. Gregory et al. take one very important step in showing this association between Neanderthal genes in modern humans, and morphological changes in the skull and brain related to such genes [5]. However, they did not test if there was any functional advantage (or disadvantage) for the perception and/or retention of visual detail and motion.

From the above, we may conclude that gene transfer from Neanderthal to modern man has resulted in more variance specifically related to visual perception in modern humans. Furthermore, during the time of contact, which may have lasted 50,000 years, it is very likely that the concentration of archaic Neanderthal genes was much higher than in modern humans, and with a wider distribution of robust genes than after a thousand generations of selection. The fact that Neanderthal remains appear in the same caves as those of sapiens sapiens, and in those caves with cave paintings, is further evidence that there was at least a period of co-habitation in the same relevant geographical areas. Hoffmann et al. recently found Iberian Cave Art of Neanderthal origin [100], as the Uranium-Thorium dating revealed the cave paintings to be older than 64.8 thousand years, which predates the arrival of modern humans in the area by at least 20 thousand years [100].

Interbreeding sapiens neanderthalensis/sapiens sapiens may have introduced hybrids that were indeed more perceptive for visual details and motion, and those individuals may have enjoyed respect in their communities, and through that respect reproductive success that favored the survival of their genes. In addition to superior visual processing the artists must have had an excellent mind-to-hand coordination, and exceptional fine-motor skills. This would make it implausible that the artists were neanderthalensis. We have found no evidence of anatomically detailed art by Homo neanderthalensis, but such art is not documented from earlier sites by Homo sapiens either (e.g., the Blombos Cave [91]. However, the Cave Art was not produced by a few talented individuals, but rather a full team of specialized workers that provided the logistics necessary to carry out extensive and complicated projects, features that we associate with culture today. The extended period of cave paintings and the associated cost of work indicate that the painters were highly respected. If selection, for this reason, favored these genes to be part of the modern genome, then we predict that modern humans with a higher proportion of Neanderthal genes will typically also show some functional advantage for processing and retaining visual details through access to alternative paths to visual perception.

## 5. Conclusions

This article has outlined much of the positive evidence for an association between visual thinkers and archaic genes, as well as associations between visual thinkers and superior visual perception and finally an association between visual thinkers and impaired language abilities. We have argued that superior visual perception is necessary for the creation of the Cave Art discussed in this article. However, positive evidence is more interesting if there is a testable prediction. As archaic genes have survived in modern man, it is to some extent testable if these genes are associated with enhanced visual perception, possibly at the expense of analytical, linguistic thinking. Recent developments have led to a feasible research program with implications for visual aesthetics as well as for the evolution of language. Our hypothesis implies that the evolution of language may hinge on a loss of an ability for visual thinking, as one hurdle towards analytical linguistic thought.

## Figures and Tables

**Figure 1 behavsci-11-00081-f001:**
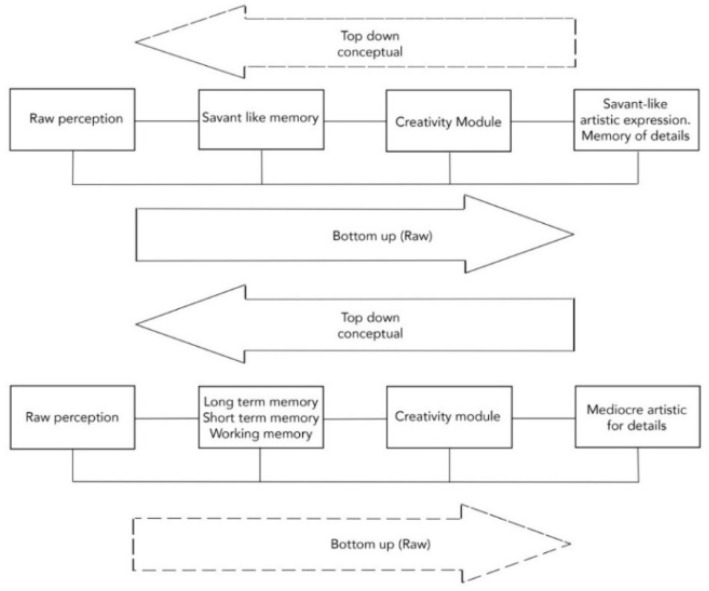
“top down”/“bottom up” directions in archaic and modern Homo sapiens.

## Data Availability

This study is based on literature review.

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
