# Peer review of "The Superior Visual Perception Hypothesis: Neuroaesthetics of Cave Art"

_behavsci, 2021, doi:10.3390/bs11060081_

Round 1

Reviewer 1 Report

The design of the paper relies on an analogical reasoning over evidence of different kinds. In short: A (savant and autistic drawing abilities) is to B (savant and autistic bottom up perceptual processing and increased visual memory) like C (UP Cave Art)) is to D (boosted creativity in drawing perceptual features and motion). On this basis, the inference is drawn that the additional property of the source domain A-B (the difference between savant/autistic and normotypes under the respect of brain areas, connection and pathways) is extended to the target domain C-D (anatomically larger brain visual areas in archaic vs. modern Homo sapiens, which granted the first ones a greater access to perceptual primary consciousness). The causal relationship for extending this property is submitted to be an imbalance of inhibitory and excitatory neurotransmitters, mediated by archaic genes that might have been preserved from archaic pre-modern non Africans ancestors to a fraction of modern population despite evolutionary pressure. 

The analogical reasoning covers so many and varied domains that the manner some basic points are framed can be improved.

1) The Authors emphasize correctly that the information processing attributed to archaic Homo sapiens caused hardly a pathology. The question remains that the limited access to generalization it implies is at odds with the ability to exploit the morphology of the cave. Being able to see the surfaces of cave walls as part of the pictures to be made requires the ability of perceiving something while recognizing it as something else. One may concede that recognition does not entail concepts, but a kind of generalization ranging over perceptual features of different kinds is implied. 

2) The aforementioned ability can figure among the cognitive markers of "behavioral modernity". The Authors quote Humphrey's claim that one must not have a modern mind to make UP cave drawings. However, they seem to suggest a more moderate view when they hint at paleoanthropological theories that support the evolutionary importance of the co-existence and interbreeding of archaic and modern Homo sapiens (p. 8, 12). Then they should spell clearly whether their reconstruction is consistent with the claim of the "human revolution" (Henshilwood; Marean), that of a multi-species model (Zilhao) or that of the accumulation and sharing of technologies and behavioral patterns across time by species in mutual adaptation (D'Errico). This point regards the adequacy of the reference literature, but even more it matters for the pivotal argument about the trade off between bottom up and top down abilities. Besides, the reader may wonder: if Neanderthals had the abilities to make drawings but also to manage blade technology, exploit aquatic resources, trade over long distances, were they behaviorally modern in some sense or was the difference between their and modern Homo sapiens’ minds not so wide? Does this issue affect the support for the Authors' claims, and if it is the case to what extent?

2.1) Since the Authors seem to emphasize the role that hybrids might have been played for the inheritance of visual abilities connected to making UP Cave figurative pictures (p. 12), a more recent reference on the Neanderthal genome could be added than the one provided in the paper (for instance Science. 2017 Nov 3; 358(6363): 655–658.) 

3) The Authors provide neither a definition nor a list of correlated features for what "raw perception" means. Does it mean visual abilities for (3.1) primitives like blobs, terminators, edges, contour discontinuities (3.2) features affecting grouping and border-assignment for figure ground structure, (3.3) primitives and features that allows perception to go from tilt and slant of surfaces to volumetric primitives and features, (3.4) all of that  provided that it is encapsulated from intentional fixation, attention and learning? Nothing more is said about the features and the neural representation of motion perception. The Authors should consider that motion perception deals with the problem of detecting a change of positions in time while preserving shape and that there is not such a thing as a sole neurobiological underpinning of motion perception (see for instance Gegenfurtner et al., Enciclopedia of Behavioral Neuroscience 2010). 

3.1) Without specification of "raw perception”, one can hardly make sense of the claim that savant brains, and allegedly archaic Homo sapiens' ones, had  no mechanism to prevent over-stimulation by "compressing" neuronal signals into a prototype. Not to mention that prototype are structures of the perceived world and basic categories may contain sensory-motors affordances.

4) The Authors conflate perceptual and figurative representational abilities. To be sure there must be a connection between them. However, lines drawn on a surface (or engraved on it) may hold different functions and stand for different features: visible or occluding edges (with different junctions), convex/concave or saddle shape surfaces, contour-endings, region boundaries. Since there is obviously no one-to-one mapping, the selection of what lines stand for requires the ability to manage the spatial properties of an array of marks and to refer by them to representational properties that display some perceptual features of the environment.

Authors should make clearer the structure of the analogical reasoning. It might be couched through another logical scheme. What matters is explaining its terms, which are actually stated only in the conclusion by referring to the independent evidence of single domains and their “association”. They should revise the manner questions, arguments and arguments are framed according to the above mentioned points, otherwise the analogical reasoning does not specify adequately the properties of the domains over which it ranges. Therefore, the validity of the inference may not be decidable by testing the set of hypotheses Authors submit at section 3.

As regards secondary literature, the reference to the MNS as the underlying neurobiological system of TOM should be corrected or, alternatively, thoroughly justified (see line 337). In fact, MNS are deemed to provide with a mind reading model that is alternative to TOM. Actually it has been primarily developed in terms of embodied simulation (Gallese Phil. Trans. Royal Society B 362(1480): 659-69). The functional interpretation of the MNS has also been taken as an alternative construal to the ventral/dorsal stream paradigm of what/where processing (Gallese Spatial Processing in Navigation, Imagery and Perception pp 329-352).

Minor language check is needed: at line 17 "modern" lacks a noun; at line 21 skull is mistyped; at lines 143-144 the sentence is difficult to read. 

Author Response

First, we would like to take the opportunity to thank the reviewers for their efforts and for valuable input.

We have done our best to make our arguments clearer, within the constraints of this article. As is so often the case with research the questions that are opened are often more interesting than the facts that have been discovered. After revision, we think that the review process has been very helpful and made some arguments clearer.

The design of the paper relies on an analogical reasoning over evidence of different kinds. In short: A (savant and autistic drawing abilities) is to B (savant and autistic bottom up perceptual processing and increased visual memory) like C (UP Cave Art)) is to D (boosted creativity in drawing perceptual features and motion). On this basis, the inference is drawn that the additional property of the source domain A-B (the difference between savant/autistic and normotypes under the respect of brain areas, connection and pathways) is extended to the target domain C-D (anatomically larger brain visual areas in archaic vs. modern Homo sapiens, which granted the first ones a greater access to perceptual primary consciousness). The causal relationship for extending this property is submitted to be an imbalance of inhibitory and excitatory neurotransmitters, mediated by archaic genes that might have been preserved from archaic pre-modern non Africans ancestors to a fraction of modern population despite evolutionary pressure. 

We thank the reviewer for these insightful suggestions. We have revised and included this in a short paragraph at the introduction, together with raising the question about the loss of superior visual perception as a price for language. It has become clearer to us that Visual Thinkers and Verbal Thinkers are two variants of thinking and they are not mutually exclusive but in competition for brain resources and also may rely on anatomically different neural pathways.

The analogical reasoning covers so many and varied domains that the manner some basic points are framed can be improved.

  • The Authors emphasize correctly that the information processing attributed to archaic Homo sapiens caused hardly a pathology. The question remains that the limited access to generalization it implies is at odds with the ability to exploit the morphology of the cave. Being able to see the surfaces of cave walls as part of the pictures to be made requires the ability of perceiving something while recognizing it as something else. One may concede that recognition does not entail concepts, but a kind of generalization ranging over perceptual features of different kinds is implied. 

This is a good observation. However, we do not claim that archaic sapiens completely lacked the ability to generalize or had very little capacity for symbolic thought. What we claim is that these painters had extraordinary perceptual abilities, seen from the perspective of modern humans. We relate this to loss of brain volumes where we assume visual processing is active, and we relate this to surviving genes, from neanderthals, associated with perception and cognition (for example AUTS2). The discussion comes later in the article, and it could be too much information to introduce when we focus on the actual artistic products.

2) The aforementioned ability can figure among the cognitive markers of "behavioral modernity". The Authors quote Humphrey's claim that one must not have a modern mind to make UP cave drawings. However, they seem to suggest a more moderate view when they hint at paleoanthropological theories that support the evolutionary importance of the co-existence and interbreeding of archaic and modern Homo sapiens (p. 8, 12). Then they should spell clearly whether their reconstruction is consistent with the claim of the "human revolution" (Henshilwood; Marean), that of a multi-species model (Zilhao) or that of the accumulation and sharing of technologies and behavioral patterns across time by species in mutual adaptation (D'Errico). This point regards the adequacy of the reference literature, but even more it matters for the pivotal argument about the trade-off between bottom-up and top-down abilities. Besides, the reader may wonder: if Neanderthals had the abilities to make drawings but also to manage blade technology, exploit aquatic resources, trade over long distances, were they behaviorally modern in some sense or was the difference between their and modern Homo sapiens’ minds not so wide? Does this issue affect the support for the Authors' claims, and if it is the case to what extent?

Henshilwood and D’Enrico have published together in many articles, and their views are likely compatible. The possibility of many ancestors for different human populations today is not a very common opinion and is not the standard hypothesis. We cannot exclude it, but in relation to Visual Thinkers we do not believe there is a need to postulate separate evolution of branches of the human tree. We are suggesting an association with some ancient genes, but we do not claim that all visual thinkers must have these genes. There could be other paths, or even mutations, that lead towards visual thinking. It could even have an epigenetic component.
   We are not trying to revise or reconstruct human ancestry, but we follow the latest information. Homo sapiens neanderthalensis is not a direct ancestor of Homo sapiens sapiens but rather a separate branch on the evolutionary tree. The branching could be so close that it is really a sub-branch within homo sapiens (as is indicated by “homo sapiens neanderthalensis”).
   However, they were close enough to produce fertile offspring (how else could some of us have their genes in our genome?). This is possibly an argument that they could belong on the same branch of the tree, where neanderthalensis is a variant of sapiens. We rely on Green et al and Gregory et al, and Pääbo in various publications for pointing out neanderthal genes in the modern human genome. It is true that the full sequencing of the neanderthal genome is in progress, and understanding the function of the variant genes is also work in progress.
    There are many recent articles, see for example Zeberg et Pääbo 2020, 2021. Since we are convinced that neanderthalensis and sapiens were genetically compatible, produced fertile offspring and both variants also produced artifacts and to some extent art, we are convinced that the two variants also had minds that were not completely incompatible. However, as pointed out by gene sequencing and differences in brain anatomy, there were definite differences in both gene variants and brain anatomy, some of variants are related to visual perception and cognition.
     The picture gets complicated as more is discovered, and there are now also other gene variants that may stem from Denisovians and even other sub-branches. This also shows how close these branches were to modern man, if also their genes have survived. Although the origins of modern humans is interesting, our reasoning mainly relates to (gene) variants with relevance for visual perception. 

2.1) Since the Authors seem to emphasize the role that hybrids might have been played for the inheritance of visual abilities connected to making UP Cave figurative pictures (p. 12), a more recent reference on the Neanderthal genome could be added than the one provided in the paper (for instance Science. 2017 Nov 3; 358(6363): 655–658.) 

This is very interesting information. Even more recently, the genome of Neanderthals is updated. We have included some recently published material with relevance for hybrids such as Zeberg and Pääbo 2020, 2021, and we expect that even more information will emerge. Still, the focus is on gene variants with implications for perception and cognition. There is, as we understand it, no doubt about the presence of hybrids.

  • The Authors provide neither a definition nor a list of correlated features for what "raw perception" means. Does it mean visual abilities for (3.1) primitives like blobs, terminators, edges, contour discontinuities (3.2) features affecting grouping and border-assignment for figure ground structure, (3.3) primitives and features that allows perception to go from tilt and slant of surfaces to volumetric primitives and features, (3.4) all of that  provided that it is encapsulated from intentional fixation, attention and learning? Nothing more is said about the features and the neural representation of motion perception. The Authors should consider that motion perception deals with the problem of detecting a change of positions in time while preserving shape and that there is not such a thing as a sole neurobiological underpinning of motion perception (see for instance Gegenfurtner et al., Enciclopedia of Behavioral Neuroscience 2010).

We thank the reviewer for the reference, which is very interesting to us for other research we are conduction on face and gaze perception (Andresen, A., Sætren, L.C., Specht, K., Skaar, Ø.O. & Reber, R. 2016a. Effects of Facial Symmetry and Gaze Direction on Perception of Social Attributes: A Study in Experimental Art History, Frontiers in Human Neuroscience, 10, September 2016, DOI: 10.3389/fnhum.2016.00452).
   For the present article, we feel that we cannot go into specifics about the detailed function of neanderthal perception (or ancient humans in general) as the genetic link to function at a neuronal level is not fully understood, and the information about brain anatomy from skull modelling can only indicate differences on a crude scale, such as volume, and mainly for features of the superficial parts of the cerebrum. There are many theoretical pathways towards extending our knowledge. We could have a microscopic theory, where we build up from fundamental features, or we could have a more phenomenological theory where we work with the available observations. We do not have the time-travelling microscope that we feel would be necessary to know in depth about the visual perceptual system of Neanderthals. However, we now know some genes and their links, and more is found out every year as more research is published on this hot topic. So with time, it is possible that we do find out more of the details, possibly through understanding the function of the gene variants in more depth, as we propose as a long term research agenda.

We have added our definition of raw perception on page 2. The main focus is that raw perception is less affected by top-down projections. We show that this can be accomplished also in modern man with drugs, transcranial stimulation, and also later in the article it is suggested that drawing techniques (and meditation) may ameliorate the effect of top-down projection, with implications for how we can recall information more accurately when we operate in modes where effects of verbal thinking is minimized. The discussion section is updated to reflect this.

3.1) Without specification of "raw perception”, one can hardly make sense of the claim that savant brains, and allegedly archaic Homo sapiens' ones, had  no mechanism to prevent over-stimulation by "compressing" neuronal signals into a prototype. Not to mention that prototype are structures of the perceived world and basic categories may contain sensory-motors affordances.

We refer to research where savant conditions can be induced, for example by transcranial stimulation. The point is that the stimulation temporarily knocks out areas that are associated with top-down projections. The argument is more about the flow of information, and involvement of higher order information, such as verbal reasoning.

  • The Authors conflate perceptual and figurative representational abilities. To be sure there must be a connection between them. However, lines drawn on a surface (or engraved on it) may hold different functions and stand for different features: visible or occluding edges (with different junctions), convex/concave or saddle shape surfaces, contour-endings, region boundaries. Since there is obviously no one-to-one mapping, the selection of what lines stand for requires the ability to manage the spatial properties of an array of marks and to refer by them to representational properties that display some perceptual features of the environment.

This is a good point. However, not all the cave paintings are engraved. The older Lion Cave shows more accurate detail than the later Sanctuary of Les Trois Fréres, Lascaux, and only the later uses the structure of the cave consistently. One point is that the older Lions Cave is less symbolic than the more recent examples. It just shows that these artists were very clever, and it shows that these activities went on for thousands of years, thus not relying on the talents of any accidental genius. We do not exclude that the painters could reason symbolically to some (large) extent, and this is also true of modern-day visual thinkers. The argument is more of a relative balance. Grandin illustrates this: she describes herself as a predominantly visual thinker, but she is capable of analytical thought to the extreme.  

Authors should make clearer the structure of the analogical reasoning. It might be couched through another logical scheme. What matters is explaining its terms, which are actually stated only in the conclusion by referring to the independent evidence of single domains and their “association”. They should revise the manner questions, arguments and arguments are framed according to the above mentioned points, otherwise the analogical reasoning does not specify adequately the properties of the domains over which it ranges. Therefore, the validity of the inference may not be decidable by testing the set of hypotheses Authors submit at section 3.

We have extended this at several places in the manuscript (all changes are marked in red). The discussion is extended with three examples where “raw perception” may interfere with verbal reasoning: Recall of word lists may be improved by drawing rather than writing (and activating verbal concepts, along with competitors), False Memories can be induced in a large proportion of the population by presenting alternative narratives that have some overlap with actual events, and Taste is affected by Color and activation of vocabulary associated with color (in wine tasting). When thinking of these examples, one of the authors recall two personal episodes. In kindergarten, more than 50 years ago, we had a small experiment of blind folded tasting and I vividly recall that I could not taste the difference between an apple and cucumber in this condition. I was also a subject in an EEG experiment on the effect of drawing on recall. I remember how excited I was when I could identify line images after just a few lines where presented. I did not reflect that the images that I recalled the fastest were those that I had previously drawn in a different condition of the experiment.

As regards secondary literature, the reference to the MNS as the underlying neurobiological system of TOM should be corrected or, alternatively, thoroughly justified (see line 337). In fact, MNS are deemed to provide with a mind reading model that is alternative to TOM. Actually it has been primarily developed in terms of embodied simulation (Gallese Phil. Trans. Royal Society B 362(1480): 659-69). The functional interpretation of the MNS has also been taken as an alternative construal to the ventral/dorsal stream paradigm of what/where processing (Gallese Spatial Processing in Navigation, Imagery and Perception pp 329-352).

We are very grateful for pointing this out. Yes, it is correct that MNS is different, and it is different in an interesting way. Both MNS and ToM are explanations of observations, but the explanation of the observations is widely different. We have tried to make this much clearer and added three references.

Minor language check is needed: at line 17 "modern" lacks a noun; at line 21 skull is mistyped; at lines 143-144 the sentence is difficult to read. 

All these have been corrected.

Reviewer 2 Report

Visual thinking and creativity in relation to Cave Art in the Upper Paleolithic  is discussed in the manuscript  “The superior visual perception hypothesis: Neuroaesthetics of Cave Art”. I found interesting information presented in the manuscript about the theories relating to the way in which the brain has developed and how that development has affected the development of art making. What was of particular interest to me is the claim made regarding that concepts are hindering finding new and better solutions and that analytic linguistic thinking is causing a decrease in visual abilities.

L 225-226

“What blocks our access to the raw perception? One explanation is, as we have seen, that inherent concepts are driving us, concepts being a hindrance for finding new and better solutions”.

L 275-279

“We find this extreme ability in chimps’ working memory a very interesting corollary to the savant brain, and a demonstration of a superior effect of raw perception that is present in a related species with a common primate ancestor, suggesting that archaic man may have had similar visual abilities, where modern man does not. Possibly this loss is a price to pay for analytic, linguistic thinking.”

L 606

“Our hypothesis implies that the evolution of language may hinge on a loss of an ability for visual thinking, as one hurdle towards analytical linguistic thought.”

One solution mentioned within the manuscript is to use drug in order to lower the control of thought processes and boost creativity:

L 426-430

“Carhart-Harris et al. ([83]) discuss access to primary consciousness, as related to the entropy of the brain. They present evidence that heightened states of creativity could be reached in modern man if using a drug (psilocybin), which elevates brain signaling and lowers the control of thought processes. Creativity could be boosted by lowering higher order control, thus decreasing conceptual processing and emphasizing low-level perception that is not normally available to our awareness; as we have seen above, this was also suggested by Gobet et al. ([42]) and others.“

I suggest looking at other options for going beyond language, lowering the control of the thought processes and boosting creativity. One example is the method of “drawing on the right side of the brain” where the students are encouraged to eliminate the words/names attached to the drawing subject (Edwards, Betty. Drawing on the Right Side of the Brain. Tarcher/Putnam, New York, 1989).

There are also new research which has shown that memory is facilitated by drawing which may be of an interest for the authors to include in the manuscript, such as:

Wammes, J. D., Meade, M. E., & Fernandes, M. A. (2016). The drawing effect: Evidence for reliable and robust memory benefits in free recall. The Quarterly Journal of Experimental Psychology, 69(9), 1752–1776. https://doi.org/10.1080/17470218.2015.1094494

Ottarsdottir, U. (2018) Processing Emotions and Memorising Coursework through Memory Drawing. ATOL: Art Therapy OnLine, 9(1). Retrieved from: http://journals.gold.ac.uk/index.php/atol/article/view/486/pdf.

At the end of the manuscript it is stated that:

L: 604-605

“Recent developments have led to a feasible research program with implications for visual aesthetics as well as for the evolution of language.”

It would have been more appropriate to include this information earlier within the manuscript and also state which research program the author is referring to. A possible avenue here would be to mention research where language and visual aesthetics is integrated through drawing. See further for example:

Ottarsdottir, U. (2018) Art therapy to Address Emotional Well-being of Children who have Experienced Stress and/or Trauma. In: A. Zubala & V. Karkou (Eds.), Arts Therapies in the Treatment of Depression: International Research in the Arts Therapies (pp. 30-47). Oxford: Routledge.

Ottarsdottir, U. (2010) Writing-images. Art Therapy, Journal of the American Art Therapy Association. 27(1), 32-39.

It would help the reader understand the subject if pictures that show the Cave paintings they are talking about each time were presented.

For example, which Cave painting is referred to here? It would help to present the image within the manuscript.

Bradshaw Foundation, especially the Lion Cave

This referencing helps: as in a bison of Chauvet ([25] (p.318,figure 2)).

L 10-11

“The normal brain function in modern man rarely supports the creation of highly detailed paintings, particularly the convincing representation of animal movement, without extensive training and access to modern technology”.

I do not totally agree that the cave paintings are such highly developed. Although, I do agree to some extend I don´t think the difference is as great as is claimed within the manuscript. I think the cave paintings are generally rather primitive and without a lot details. There is for example not a lot of three-dimensional quality for Cave paintings in comparison to modern art making. 

Artists in general have protected their visual thinking and creativity more so than other modern men. Through integrating drawing and art making to a larger extend into education and other labour this quality would most likely be facilitated to a greater extend within the modern man.

A space is needed between “A” and “more” in this sentence:

L 62-63

Amore information-rich perception and detailed memory puts higher demands on raw processing ability, which is consonant with larger brain volumes in archaic Homo sapiens (cf. [9]).

L 77-7

This sentence is difficult to comprehend:

“A person not only does not need a typical modern mind to draw like that (in the caves) but must not have a typical modern mind to draw like that.“

Although I find the neuroaesthetics introduced within the manuscript interesting I´m not a specialist in neurology of the brain and therefore I will not comment on that part. However, I found the focus of the manuscript somewhat deluded and was not sure why to include some parts and how they related to the main theme of the article. For example regarding mirroring and thee mirror neuron system:

L 334-343

“The autistic brain shows significant differences for ToM ([67] inter al.), and recent 334 research has demonstrated that people with ASD have a significantly reduced activation in the ToM associated brain areas in response to direct gaze ([68]) compared to normal controls. ToM depends on a normal functioning mirror neuron system (MNS). Individuals with ASD suffer a dysfunction of their MNS, impeding the sensory/motor and the emotional mirroring mechanisms, preventing a person’s ability to understand other people’s 339 emotions, resulting in lack of empathy ([69, 70]). This means that the normal understanding of the intention intuitively attributed to gestures and face expressions is more or less absent in people with ASD. In general, social relationships alters in persons with ASD, and “ [s]ocial consciousness is assumed to engage cortical areas, including the superior temporal-sulcus, the temporoparietal junction, and the medial PFC (prefrontal cortex), mostly in the right hemisphere” [71] (p.176).“

I suggest going through the text and investigating how each paragraph relates to the main topic of discussion and focus of the manuscript.

Author Response

We thank the reviewer for so many positive remarks and good input.

We have updated the manuscript, and introduce the idea related to a cost for language earlier in the manuscript and the discussion is also updated with reference to three examples were visual thinking and verbal reason may interfere for recall.

Reflecting on raw perception and the interference with language made me recall two episodes in my own life. In kindergarten, more than 50 years ago, we had a small experiment of blind folded tasting and I vividly recall that I could not taste the difference between an apple and cucumber in this condition. I was also a subject in an EEG experiment on the effect of drawing on recall. I remember how excited I was when I could identify line images after just a few lines where presented. I did not reflect that the images that I recalled the fastest were those that I had previously drawn in a different condition of the experiment.

The discussion is extended with three examples where “raw perception” may interfere with verbal reasoning: Recall of word lists may be improved by drawing rather than writing (and activating verbal concepts, along with competitors – the Wammes et al article), False Memories can be induced in a large proportion of the population by presenting alternative narratives that have some overlap with actual events, and Taste is affected by Color and activation of vocabulary associated with color (in wine tasting).

We have gone through the manuscript and tried to clarify the argumentation and improve the flow of reading.

All changes are marked in red.